# Why women with breast cancer presented late to health care facility in North-west Ethiopia? A qualitative study

**Aragaw Tesfaw**[1]*, **Wubet Alebachew**[2], **Mulu Tiruneh**[1]

1 Department of Public Health, College of Health Science, Debre Tabor University, Debre Tabor, Ethiopia,
2 Department of Nursing, College of Health Science, Debre Tabor University, Debre Tabor, Ethiopia

* aragetesfa05@gmail.com

**Data Availability Statement:** All relevant data are within the paper and its Supporting Information files.

**Funding:** The author(s) received no specific funding for this work.

## Abstract

### Background

Although early diagnosis is a key determinant factor for breast cancer survival, delay in presentation and advanced stage diagnosis are common challenges in low and middle income countries including Ethiopia. Long patient delays in presentation to health facility and advanced stage diagnosis are common features in breast cancer care in Ethiopia but the reasons for patient delays are not well explored in the country. Therefore we aimed to explore the reasons for patient delay in seeking early medical care for breast cancer in North-west Ethiopia.

### Methods

A qualitative study was conducted from November to December 2019 using in-depth interviews from newly diagnosed breast cancer patients in the two comprehensive specialized hospitals in North West Ethiopia. Verbal informed consent was taken from each participant before interviews. A thematic content analysis was performed using Open Code software version 4.02.

### Results

Lack of knowledge and awareness about breast cancer, cultural and religious beliefs, economic hardships, lack of health care and transportation access, fear of surgical procedures and lack of trusts on medical care were the major reasons for late presentation of breast cancer identified from the patient's narratives.

### Conclusions

The reasons for late presentation of patients to seek early medical care for breast cancer had multidimensional nature in Northwest Ethiopia. Health education and promotion programs about breast cancer should be designed to increase public awareness to facilitate early detection of cases before advancement on the existing health care delivery system.

**Competing interests:** The authors have declared that no competing interests exist.

# Background

Breast cancer is a group of diseases which results from uncontrolled growth and changes on breast tissue typically resulting in a lump or mass. It is the 2nd leading cause of cancer among females worldwide and a growing public health burden in low-resource settings [1, 2]. Based on the *GLOBOCAN estimates* around 2.1 million female breast cancer cases were diagnosed in 2018, accounting for almost 1 in 4 cancer cases among women [3]. It is the most prevalent cancer in African women, although its incidence is lower compared to high-income countries [4]. Approximately 1 in 8 women (13%) will be diagnosed with invasive breast cancer in their lifetime and 1 in 39 women (3%) will die from breast cancer [1].

As of World Health Organization (WHO) 2018 cancer country profile report, breast cancer is the leading cancer in Ethiopia with the highest age standardized mortality rate of 22.9 per 100,000 population [5]. Similarly according to the Addis Ababa population-based cancer registry report, breast cancer becomes the most frequently diagnosed malignant tumour accounting 33% of cancer cases in women and it is the commonest cancer in four of the six Ethiopian regions. The Estimated national Age Standardized Incidence rate for women was about 43 per 100,000 population [6].

As evidence shows an important determinant factor for breast cancer survival is the degree to which cancers are detected at early stages while advanced stage and large tumor size at diagnosis are associated with decreased survival and worse clinical outcomes [7, 8]. Localized disease have higher five year survival (99%) than metastasized or advanced stage cancer cases (26%) [9]. Similarly patients diagnosed at early stage of breast cancer showed better survival than patients with late stage in Ethiopia [10].

Studies showed that breast cancer is often diagnosed at an early stage and patients have good prognosis in developed countries. However, it is more often diagnosed at advanced stage and patients have low survival rates in developing countries [11]. Nearly three fourth (72.5%) of breast cancer patients diagnosed at advanced stage in Ethiopia [12].

An evidence from systematic review showed that patient delays in seeking early medical care for breast cancer is a major contributor for advanced stage diagnosis of breast cancer among African women [13]. Other studies also reported that patient delays in diagnosis affect cancer stage at diagnosis even though their characteristics is vary in particular countries [14–16]. The way in which individuals understand and perceived their initial breast symptoms had shown to influence early medical seeking behavior due to their misunderstandings about the first clinical symptoms of the disease [17]. Patients who had painful breast wound were more likely to be presented early as compared with patients who had painless lumps [12, 14].

Lack of knowledge and awareness about breast cancer is found to be a contributing factor for late presentation to breast cancer care in most sub-Saharan African countries [13, 18].

The Breast Health Global Initiative (BHGI) recommends on to increase public awareness about risk factors, initial symptoms and early detection methods of breast cancer as well as providing accessible and effective diagnosis services for down staging breast cancer [19]. However there are still very few initiatives in developing countries to address the issue [20]. In Ethiopia, women's knowledge about risk factors for breast cancer is low [21].

Breast cancer incidence is rising and becoming a major public health problem in Ethiopia which poses a substantial threat in the country with limited oncology centers [22, 23]. A large proportion of breast cancer patients in Ethiopia present for medical care too late, or not at all, resulting in high mortality [24]. The overall breast cancer screening practice is very low in the country (9%) [25] only few breast cancer patients sought medical advice within the first year (average time to presentation to health care facility of 1.5 years from their initial symptom recognition) [26]. Knowledge about breast cancer early detection methods (breast self-

examination and clinical breast examination) is low in the country [27] only less than half (32.5%) of women practiced breast self-examination [28]. In addition women's knowledge about risk factors for breast cancer is low [21]. About 73% of the patients had patient delays s of >90 days with a median presentation delay of 4 month and this patient presentation delay more than three month is associated to advanced stage of the disease [12]. Patient related barriers were one of key reasons for early diagnosis of breast cancer in south and southwestern Ethiopia [29].

Although Ethiopia has been developed the National Cancer Control Plan in 2015 for improving early diagnosis of cancer, and efforts are made, more than two-thirds of breast cancer cases are diagnosed at advanced stages of the disease and need palliative care [30]. There is only one radiotherapy center (Addis Ababa, Tikur Anbesa Hospital) in the country. Although some studies are done at the center and southwestern part of the country, little is known about the reasons for late presentation in North western part of the country. Therefore exploring the major reasons for women for delay in seeking medical care immediately when they saw breast cancer initial symptoms is crucial for designing prevention strategies at local and national level.

### Clinical implication of the study

This study answered the question of why women with breast cancer presented late to health care facility in North West Ethiopia which is crucial particularly for health care providers and policy makers to develop specific health education and promotion interventions to promote early detection of cases before advancement at each level of health care delivery system in the country. The result will also insight health care providers to encourage women to regularly check their breast, to come for clinical check-ups and they will do clinical breast examination for all women coming to health care facility for other medical visits like anti-natal care, postnatal care follow ups, family planning purposes.

## Methods

### Study area and approach

A qualitative study using phenomenological approach was used to explore the reasons for late presentation of women for breast cancer care. We used these method since it is important to study "lived experiences of individuals about a certain phenomenon" in this case, the experience of breast cancer patients with their initial symptoms, their reasons for delay for seeking medical visit [31]. The study was conducted from November to December 2019 in two comprehensive specialized teaching hospitals located in the North-western Ethiopia (University of Gondar and Felege Hiwot Specialized hospitals). The hospitals are used as the only oncology referral centers for all cancer cases including breast cancer in the Amhara Regional State, Ethiopia. Felege Hiwot hospital is found in Bahirdar city 565 km far from Addis Ababa (the capital city of Ethiopia) while University of Gondar specialized hospital is a comprehensive specialized and teaching hospital found 724.7 km far from the capital city of Ethiopia (Addis Ababa). The hospitals play an important role in teaching medical and other health science students in different streams and specialities. The hospitals currently provide diagnostic, surgical and chemotherapy treatment service for cancer patients including breast cancer.

### Participants of the in-depth interview and recruitment procedures

A total of 14 In-depth Interviews (IDI) were conducted in the two oncology units. The participants in this study were newly diagnosed female breast cancer patients in the two specialized

hospitals (University of Gondar and Felege Hiwot Specialized hospitals). The hospitals (above) were selected because they are the only oncology centers in the region and serve mostly rural populations and have experience with treatment services for breast cancer patients. The investigators communicated first the hospital director and the oncology staffs and provided description of the study. Participants were selected using purposeful sampling technique [32]. Women whose age greater than 18 years and who were diagnosed with breast cancer from November to December 2019 and treated at the two oncology clinics were included in the study. The sample size was determined based on theoretical saturation of data during the time of data collection [33].

### Data collection procedure

Participants of the study were selected with the help of the oncology unit nurses. After we obtained informed consent to participate, an appropriate place and time for an interview was arranged by the oncology unit coordinators. Interviews were conducted face to face with participants in a private room at the oncology unit. Two public health professionals who had a Master of public health in health education and promotion collected the data with the assistance of two note takers. Interviewers were fluent in written and spoken Amharic and English language.

A semi-structured interview guide was used to conduct a face to face in-depth interview with breast cancer patients (see S1 File). The guide was developed thorough literature review. It was first developed in English and then translated to the local language (Amharic) to facilitate discussion with patients. The discussion was focused on reasons for late presentation (why women's did not immediately visit health facility when they encountered any breast abnormalities or changes in their breast). The interview guides consisted of open-ended questions like what do you understand about breast cancer before? What did you feel when you notice changes in your breast? How long did you live with the signs or symptoms before you visit health facility? Interviewers also used probing questions which guided the patients during the interview to get further clarification of the responses of the participants. All of the interviews were audio-recorded. Field notes were taken during the in-depth interview to include keynotes of the participants and to use as a backup in case the video records are lost.

The in-depth interview was conducted on 14 newly diagnosed breast cancer patients who had self-reported delay of more than three month from their initial symptom recognition until their first health care visit. Six of the patients were from University of Gondar and eight were from Felege Hiwot specialized hospitals. Patients were newly diagnosed with breast cancer during data collection period and they were recruited from oncology unit at selected hospitals.

Participants were encouraged by different techniques like probing of the questions to talk more about all aspects of the reasons for their delay in presentation to health facility. Demographic and clinical data was collected through medical records and from the participants during the interview.

All interviews were audio-recorded after the verbal consent of the study participants. The interview was continued until a point of saturation of information was reached. Each interview had taken approximately 40–60 minutes with the average interview time of 46.67 minutes.

The transferability of the findings was ensured by collecting data on two oncology units which are used as referral centres for cancer patients in North West part of the country (Amhara Region). This helped to get women from different corners of the region with different socio-demographic and cultural background. The data collectors and investigators were met on a daily basis after data collection and debriefed on their daily interview findings.

To maintain the trust worthiness and validity of the findings, the investigators developed rapport with patients who were participated in the study. Credibility was maintained through

participant checking during in-depth interview, and through feedback of findings at the end of the study from whom the data was taken. Keeping a diary with information about impressions was enhanced conformability. The dependability of data was maintained by taking depth of information until reaching data saturation point which reflects a sample needed for the study i.e. when repeated patterns become apparent in the patients' narrations.

### Data analysis procedure

The qualitative data generated from in-depth interviews (audio recorded data) was transcribed verbatim by re-listening the tape recorder several times and reading the field notes line by line and translated from Amharic to English language. The transcripts were cross-checked for any mistakes by simultaneous reading of the transcripts and listening the audio-recorded voices. Transcripts and translations were cross-checked for accuracy and consistency by two independent persons. Verbatim transcripts were imported into Open code software version 4.02 for coding similar ideas together without compromising the central idea. The coding was done line-by-line by the principal investigator. First, repeated responses from each question was identified then similar responses were grouped into codes, similar codes were categorized and similar categories were grouped in two themes. Thematic content analysis were used to analyse the data. The most frequently said quotes of respondents were selected and presented in italics. Where necessary, translated English grammar in direct quotes was corrected from transcriptions to ease readability and where necessary, placed in the first person context, but the content remains unchanged. The investigators undertook the entire research process from patient selection to publication of the results.

### Ethical consideration

An ethical clearance letter was obtained from a Research Ethics Committee of Debre Tabor University, College of health sciences (reference number: 782/2012 E.C. /CHS). The permission and agreement consent was obtained from the study hospitals prior to the study after a brief explanation of the purpose of the study through support letter. Informed verbal consent was obtained from all participants of in-depth interview after a brief explanation of the aim of study. Confidentiality of information and privacy of participants' interview was respected. The participants were told that information they provide use only for the purpose of this study. The names of the participants did not included in the interview guide rather specific codes were used. At the end of each interview necessary advice and right information, concerning breast cancer was provided to patients.

## Result

### Socio-demographic and clinical characteristics of breast cancer patients

The table below describes the socio-demographic characteristics of breast cancer patients participated in the in depth interview. A total of 14 newly diagnosed breast cancer patients were participated on the in-depth interview. The age of patients were ranged from 27 to 68 years with the median age of patients at diagnosis was 43.5 years. The majority 9 (64.3%) of the patients were diagnosed at stage III. Majority 12(85.7%) of the patients have delay of more than six month from their initial visit (Table 1).

### Patient reasons for late presentation for breast cancer care in Northwest Ethiopia

Sociodemographic, cultural, religious, access to health facility and transportation related reasons were explored from the narrations of breast cancer patients for delay presentation to

**Table 1. Socio-demographic characteristics of breast cancer patients diagnosed at University of Gondar and Felege Hiwot Specialized hospitals North-west Ethiopia, 2019.**

| Characteristics | Frequency | Percentage |
|---|---|---|
| Age group | | |
| <40 | 6 | 42.8 |
| ≥40 | 8 | 57.2 |
| Home residence | | |
| Rural | 10 | 71.4 |
| Urban | 4 | 28.6 |
| Educational status | | |
| Illiterate | 5 | 35.7 |
| primary education completed | 4 | 28.6 |
| secondary education completed | 3 | 21.4 |
| College and above completed | 2 | 14.3 |
| Religion | | |
| Orthodox | 12 | 85.7 |
| Muslim | 2 | 14.3 |
| Marital status | | |
| Married | 11 | 78.6 |
| Single | 3 | 21.4 |
| Occupational status | | |
| Farmer | 7 | 50 |
| Government employee | 2 | 14.3 |
| House wife | 5 | 35.7 |
| Stage of breast cancer | | |
| Stage II | 5 | 35.7 |
| Stage III | 9 | 64.3 |
| Time from to first visit (patient delay | | |
| <6 month | 2 | 14.3 |
| >6 month | 12 | 85.7 |

medical care. Themes and subthemes were emerged from the narrations of the participants regarding the reasons for late presentation in seeking medical care for breast cancer (see S2 File).

## Lack of knowledge and awareness related reasons for late presentations to breast cancer care

As the in-depth interview participants narrated, one of the major reason for late presentation for breast cancer care was lack of knowledge and awareness about breast cancer risk factors, early symptoms, early detection methods and treatments. Almost all patients did not know about breast cancer before their diagnosis. As the participants mentioned they were not aware about initial breast cancer symptoms, risk factors of the disease and diagnostic methods consequently they could not recognize the early breast abnormalities and did not want to seek early medical care. As the participants explained, the community as well as their family members did not know about breast cancer risk factors, and early detection methods rather link such kind of problems with other medical problems. Almost all breast cancer patients participated in this study were associate their initial breast symptoms with other problems and some considered it as a simple and self-limited condition. The lack of knowledge and awareness about

breast cancer combined with the different perception of patients and community contributed for their late presentation to health care services consequently, to be diagnosed at advanced stage of the disease.

> "...I did not heard about breast cancer. I saw the changes on my breast while I was sleep once up one a time but I did not think that it will be this kind of disease...I knew a women ((my husband's sister) who had similar problem with me but her was healed by itself." **(A 64 years old patient)**

> "...I did not know it before and I did not heard the name itself. The society said me it is 'MITCH (a local name given for any breast swelling in which the community perceived as it is due to exposing of breast to sunlight)... no one educate me about breast cancer before." **(A 58 year's old patient said)**

## Initial symptom misinterpretation and poor practice of early detection methods

All participants explained as there initial symptom was painless swelling on their breast. As they described they were not worry about it at first notice but the swelling was gradually increase in size and started pain which alerts them to seek care. Most of them detect the abnormalities unintentionally during take-off their cloths or when taking shower. Most of the patents did not regularly practice self-breast examination even they do not know about practice of breast self-examination. All did not have history of clinical check-ups for their breast. All women did not have history of mammography check-ups.

> "No, No. I did not have experience of checking my breast. I saw the swelling when I was washing my body at river...I did not do breast self-examination before." **(A 27 years old patient)**

> "...ehhhh...I did not examine my breast. I saw the swelling by accident... I was not going to health facility for check-up of my breast." **(A 42-year-old patient said)**

The other important reason explained by breast cancer patients for delay in seeking care for breast cancer was their misunderstanding of early symptoms of the disease. Though nearly all women first noticed lumps, all did not sought medical advice early, most of them started to seek medical consultation after 6 months of initial symptom recognition. As they clarified changes in their symptoms mainly the pain and enlargement of the swelling driven them to seek medical advice. Most participants did not think the initial lump would be cancer. As the patients reflected most of them it as easy and self-limited condition. As they said they waited triggering factors like pain, wound, pus or discharge to go to health care facility. As a result they fail to visit medical care early unless such things become manifested.

> "... first it was a very small swelling and it was painless so I was not consider it as it will be sever disease but the wound becomes burst and tried to produce discharges...emmm... if I knew about cancer, I will come early when I saw the swelling early in my breast. But I ignored the initial swelling..." **(A 50-year-old patient said)**

> "...my problem last a long time. I felt a painless swelling on my breast just before 3 years but I stayed healthy and it was not have any pain. So I was not think that it could be serious and severe my family also did not think that it will be cancer, they told me as it is self-limited and held by itself..." **(A 38 year's old patient said)**

## Access to health care facility and transportation related problems

The patients had faced problems to access health facilities on time due to long distances and high costs of transportation. Majority (10 of the participants) of the participants were from rural residence and faced such difficulties when accessing medical care. As the participants reflected most of the rural women came late to health facility after the disease is advanced. As the patients said patients from rural or hard to reach areas are not usually visited by health extension worker as a result they could not get education about cancer and other health related issues. Because of these their knowledge about the disease is low.

> *"I am not educated. I lived in the rural area. . . .I am a farmer. My breast problem starts before two years. . .The road is not safe for transportation from my home to health center so most of the time we go through foot to get the nearby health center."* (A 48 year's old patient said)

## Cultural and spiritual related reasons for late presentation to breast cancer care

As narrations from the participants indicted cultural and religious perceptions and thoughts had their own contributions for their late medical care seeking for breast cancer. As the participants described almost all patients delayed seeking medical care just because they were spent time by using traditional treatments and spiritual treatments options. Almost all of the patients were used holy water and herbal medications before they visited medical care. In addition most of the patients have a strong belief on use of herbal medications and spiritual treatment options for treating anybody swellings than the modern medical care. Some patients also perceived that cancer is a disease due to sin and they consider it as GOD's penalty. As the patients narrated almost all women came to health facility when they lose hope after they tried all spiritual and herbal treatment options. Most patients came to health facility as a last option. As the participants described most patient's believed cancer as a cultural diseases resulted from either due to sin of the individual or just by GODs punishment as a result they took herbal medication which prepared from green leafy vegetables, other plant roots and leaves by the herbalists in the community. Additionally the community perceived as if they go to health facility and get injection, the needle will make it to spread inside the body and kill the patient.

> *"I think this is the disease from GOD. . . ...I saw a small swelling on my right breast but gradually it increases and becomes large. When I follow it becomes grow then I fear and I was going to traditional healer and I applied herbal medication on it. But it does not improved me."* (**A 29 year's old patient**)

> "*. . .when I saw the swelling in my breast, I became worried and I went to church for pray and took holy water* for a long time. *I came know to hospital since the discharge become offensive. I could not sit near to other people because of the smell of the wound".* (**A 27 year's old patient**)

> "*. . .When I followed, the swelling becomes increase in size then I feared and I went to traditional healer and he applied herbal medication on it (KEBAW). But it does not improved me rather the swelling becomes burst".*" A 37 years old patient

## Fear of surgical procedures and lack of trust on medical care

As the breast cancer patients explained most of them believed that cancer cannot be treated medically and they perceived as any type of cancers do not have any medical treatment rather they trust on use of herbal medications and spiritual options like pray and holy water. Because

of these most of the patients first visit traditional healers and spiritual areas and finally they tried to visit health care facility as a last option while some others lose hope and prefer dying in their home. As the women's explained in addition to lack of trust in medical care, they fear the medical procedures which will be done for patients. As they said there is also a strong perception in the community that if a women go to hospital, they think as her breast will be cut and she will be died. The other important point mentioned by the participants was the community believe that the women's breast is surgically removed, they do not perceived that she will be survived. The other barrier mentioned by the patients was they perceived that once a women's breast is removed, she could not give birth and she could not married. They think as the women lost her feminist character. As most of the women's said as the community advised them as not to go to health facility rather they advised them to die without losing their breast. As they said the community perceived it as a dignity. Because of the above mentioned factors most women do not seek early medical care and usually presented at advanced stage of the disease.

> "...No one of my family member allowed me to go to operation and to remove my breast ...They do not trust medical care even they believed that you will be died if you go to hospital and operated. But thy mentioned the experience of some other patients who died after operation and some others who cured applying herbal medication" **(A 36 years old patient)**

> "... I know two patents who were diagnosed with breast cancer and whose breast was removed but after some years they died so the community knows this issue and they did not have a hope on medical treatment." **(A 32 years old patient)**

> "... I am very aged and my neighbours were advised me to not to undergo surgery for my breast rather they advised me to die without losing it." **(A 68 years old patient)**

## Economic hardships and lack of support

Economic problems and social support were mentioned as reasons for late presentation to breast cancer care. As the participants explained most patients did not have adequate money for transportation costs. Those individuals who have money will go to the nearby health facility early for medical check-ups and they will get early diagnosis and treatment but others will stay at home a long time to find money so that they could not seek early medical care at a health facility. As the patients described most of them faced difficulty of getting money for their medical care and transportation costs. Patients also described that they lost some of their time by collecting money for their expense so they wait until that money is collected. This contributed for their delay in visiting health facility early when they saw the changes in their breast. Most of the women also described as they did not have any family and social support.

> "... No one support me economically and I did not have adequate income even to grow my children's. As a result I could not come early to hospital. I lived with the swelling more than two years since I could not afford the cost for diagnosis and treatment." **(A 45 years old patient).**

> "…. I am a farmer and our harvesting system is not satisfactory. We live just hand to mouth. I do not have any money for further treatment after now…. I have no any family who can support me." **(A 42 years old patient)**

## Discussion

Early detection of breast cancer is an important determinant factor for improving prognosis and to decrease mortality form the disease however delays in diagnosis and treatment results

incurable disease and low survival rates [10, 11]. Breast cancer patients mainly from developing countries usually present late to health care facilities due to a number of reasons and usually diagnosed at advanced stages [34]. In contrast most of the patients from high income countries presented early to medical care and breast cancer is detected at earlier stages which results lower mortality and high survival rates [35]. So exploration using qualitative approach is an appropriate for an in-depth understanding of opinions, thoughts and feelings of breast cancer patients, and obtaining more detail information on the reasons for their delay in seeking early medical care.

Different studies reported several patient related reasons for delay for seeking early medical for breast cancer care however each of the reasons varies across the regions depending on the awareness, perception, accessibility of health infrastructure and economic status of countries [13, 36]. In this study we tried to explore the reasons for late presentation of breast cancer patients in seeking early medical care. Our study explored socio-cultural, religious and economics related reasons as the major reasons for patient delay to seek care for breast cancer. Lack of knowledge and awareness about breast cancer risk factors, initial symptoms and early detection methods, belief in herbal and spiritual options and economic hardships, lack of access for transportation and health care facilities, and belief that cancer has not any medical treatment were the major reasons identified for late presentation of breast cancer patients.

Lack of knowledge and awareness about breast cancer is the most important and preceding reason for late presentation to breast cancer care in our study. This finding is in line with the findings from other studies in most developing countries which stated poor knowledge and awareness was a major contributor for very long delays at home before seeking early medical care in most breast cancer patients [13, 20]. This study found that a strong interplay between the perception and early medical care seeking for breast cancer care. The participant's knowledge and perception about breast cancer symptoms and risk factors is very low. Patients consider themselves not susceptible to the disease, although some believe that the condition would have potentially serious consequences as a result they are less likely to seek early medical care for any changes on their breast. This finding is similar to findings from other studies conducted in southwest Ethiopia and Addis Ababa (capital city of Ethiopia) [29, 37].

Most patients believe that breast cancer is a disease due to GOD punishments as result of their misconduct of religious rules or due to their sin and they think that the disease did not have any medical treatment. This misconception might also be the result of very low awareness and a sense of not being vulnerable. The result is in line with other qualitative findings in Ethiopia and other African countries [29, 38, 39].

In our study, participants believed that breast cancer is a serious and deadly disease. As the narrations from the patients showed that breast cancer is the most feared disease in the community but knowledge about the risk factors, and medical treatment options is low rather high trust on religious and traditional treatment options. The result is consistent with the study conducted in Addis Ababa and other countries [13, 29, 37].

In this study, the accessibility to media and health education and promotion services about breast cancer is limited mainly to rural people. The women's knowledge and practice about breast self- examination is very poor. Despite the misconceptions related to the cause of breast cancer and its screening, the pattern of early medical seeking behaviour is low. A number of socio-cultural and religious barriers were mentioned that influence early medical seeking behaviour which is similar to other study findings [40, 41].

A major important patient related reason for seeking early medical care for breast cancer was the low access to information about breast cancer mainly for the rural populations. This reason was explained by the patients as their long distance from health care facilities and inaccessible to Medias particularly for the rural people makes them to have low information about

breast cancer. Lack of knowledge about the disease delays in the search for early medical care, even when patients have seen the early signs and symptoms of the disease, they do not want to go to health facility since they perceived as their symptoms will be self-limited and not as such serious disease. This finding is similar to studies in other Sub-Saharan African countries [11, 13].

The findings from our study showed that most breast cancer patient's delay in seeking medical care since they wait until their sign and symptoms become worse. This is mainly related with traditional or religious practices, which are strongly practiced in most Ethiopia populations [42]. Most breast cancer patients prefer to use herbal medications and holy water before they go to health care facilities following this most patients presented to hospital after trying all of these cultural and religious treatment methods which makes them to lost their time and suffer very long delays and consequently presented at advanced stages. This finding is similar to findings from southwest Ethiopia [29].

Finding from this study revealed that patients' fear of medical procedures including mastectomy contributed for their delay to seek early medical care for their breast problem. This is associated with the community's perception and fear that if the women is undergo surgery, she will be died and could not give birth without breast. In addition it is not acceptable in some communities to have one breast so women's fear of this social stigma and discrimination since they believe that if a women losses her breast, she cannot give birth. This finding is consistent with other qualitative studies conducted at the capital and southern Ethiopia [29, 37].

The other important reason mentioned for presentation delay was being far from health care facilities mainly for the rural community in which they faced difficulties to get health education other information's from health care provider's advice and education about cancer. Patients from rural areas describe as they suffer long delays in presentation to health facilities associated with long distance to cancer diagnostic centres and lack of information access. This is similar to findings from other studies in which being from rural areas are experienced long delays in presentation to health facilities compared with urban people [14, 17]. Most the rural people are far from health care facilities and lack of access to information through leaflets, newspapers and other media's because of this they have low knowledge and awareness about the disease and usually do not want to go to health care facilities early unless the disease becomes sever and sever.

Patients explained that the other major reason contributed for their delay in seeking early medical care is the way they give meaning for their early breast signs and symptoms. Most patients relate their first breast symptoms with other medical conditions like breast feeding, use of contraceptives and other disorders. As a result they delay in visiting health facilities as early as possible. They usually visit health care facilities when the initial symptoms become painful and when it affects their daily lives. This demonstrates poor awareness and knowledge of patients regarding importance of these early warning signs and symptoms of breast cancer. It was also explained as that painless swellings were often considered not serious or self-limiting. Similarly in Egypt breast cancer patients who did not have pain were present at later stage than those having pain as the initial symptom [43].

Majority of women thought that breast cancer is caused by a sin (supernatural power). Most patients fear socio-cultural stigma and discrimination. This is similar to a study conducted among reproductive age women in Ethiopia [40].

Clearly, in our study lumps were the first noticeable signs of breast cancer typically recognized by patients. Almost all of the patients in this study noticed a lump at some point accidently and most participants also misinterpreted it at first, as nothing to be serious. In fact, most patients ignored their lump for several years by relating it with other benign medical conditions. Most patients noticed more lumps or changes in their breast (pain, itching), which

triggered them to seek advice, from herbalist, clinic, or other religious means. However in some breast cancer patients their cancer was discovered when they went to health facility for other medical problem. Several other studies have similarly found that lumps are the dominant symptom noticed by women with breast cancer and that most women find lumps as their primary symptom. Studies also indicate that women in low-resource areas delay seeking care longer than women in other parts of the world, with delays of a year or more from detection of symptom to seeking advice [12, 26, 29, 38]. Most women with breast cancer in African face significant delays in accessing care through overburdened health care systems and with limited resources; adding more than a year of delay from noticing a symptom to action increases the chances that their disease will progress significantly before care initiates [13, 18]. Participants' awareness about the causes, risk, early symptoms, early detection methods, and treatment of breast cancer were poor and patients have a sense of hopelessness and uncertainty about the effectiveness of medical care rather applying spiritual actions (using holy water, pray) or seeking care from traditional healers were repeated responses from the in-depth interviews. The findings are similar to a study other parts of Ethiopia [12, 29, 37].

## Strength and limitation of the study

Our study has certain strengths and limitations. To the best of the researcher knowledge, this is the first qualitative study conducted in North western part of Ethiopia to explore the reasons for late presentation of women to seek medical care for breast cancer. However, we only used in-depth interviews for data collection with only from patient's perspective which might be better if it involves focus group discussions from health care providers and community key informants (HEW's, religious leaders, and community health agents).

## Conclusion

In conclusion, Lack of knowledge and awareness about breast cancer, initial symptom misinterpretation and poor practice of early detection methods, strong reliance on traditional and spiritual means of treatments, economic hardships, lack of health care access and transportation problems were the major reasons for late presentation of breast cancer patients. This shows the need to design intensive public campaigns to increase public awareness on breast cancer signs, symptoms, and treatment options and strengthening the capacity of the health care system is also essential to tackle the problem through early detection of cases and efforts to downstage the clinical presentation of breast cancer at local and national health care delivery levels.

## Supporting information

**S1 File. In-depth interview guide.**
(PDF)

**S2 File. Themes, categories and codes identified from in-depth interviews.**
(PDF)

## Acknowledgments

We would like to acknowledge the cooperation of each of the oncology unit staffs and medical directors of the study hospitals, data collectors and all participants in the interviews.

## Author Contributions

**Conceptualization:** Aragaw Tesfaw.

**Data curation:** Aragaw Tesfaw, Wubet Alebachew, Mulu Tiruneh.

**Formal analysis:** Aragaw Tesfaw.

**Methodology:** Aragaw Tesfaw.

**Software:** Aragaw Tesfaw, Mulu Tiruneh.

**Supervision:** Wubet Alebachew.

**Writing – original draft:** Aragaw Tesfaw, Wubet Alebachew.

**Writing – review & editing:** Aragaw Tesfaw, Wubet Alebachew, Mulu Tiruneh.

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
