## [Decision Letter · Decision Letter 0]

11 Sep 2020

PONE-D-20-18576

Why women with breast cancer presented late to health care facility in North-western Ethiopia? A qualitative study

PLOS ONE

Dear Dr. Tesfaw,

Thank you for submitting your manuscript to PLOS ONE. After careful consideration, we feel that it has merit but does not fully meet PLOS ONE’s publication criteria as it currently stands. Therefore, we invite you to submit a revised version of the manuscript that addresses the points raised during the review process.

We look forward to receiving your revised manuscript.

Kind regards,

Alvaro Galli

Academic Editor

PLOS ONE

Journal Requirements:

2.  Please include a copy of the interview guide used in the study, in both the original language and English, as Supporting Information, or include a citation if it has been published previously."

3. Please amend your current ethics statement to address the following concerns: Please explain why written consent was not obtained, how you recorded/documented participant consent, and if the ethics committees/IRBs approved this consent procedure.

Reviewers' comments:

Reviewer's Responses to Questions

**Comments to the Author**

1. Is the manuscript technically sound, and do the data support the conclusions?

Reviewer #1: Partly

Reviewer #2: No

2. Has the statistical analysis been performed appropriately and rigorously? 

Reviewer #1: No

Reviewer #2: N/A

3. Have the authors made all data underlying the findings in their manuscript fully available?

Reviewer #1: No

Reviewer #2: No

4. Is the manuscript presented in an intelligible fashion and written in standard English?

Reviewer #1: No

Reviewer #2: No

5. Review Comments to the Author

Reviewer #1: It is a very good effort to describe the reasons underlying late presentation of breast cancer in North Western Ethiopia. Some genuine reasons have been presented as to why the breast cancer patients approach late to the health care facilities in that part of the country. However, the data of only 14 breast cancer patients spanned over the record of 1-2 months is not insufficient for any statistically significant information. I would suggest the authors to include as much number of patients as possible to report a significant data for such type of study.

Reviewer #2: Background:

1. This is a qualitative research design that examined the experiences of 14 women who delayed seeking medical care for breast cancer symptoms and were diagnosed with advanced breast cancer. I think the manuscript would be more interesting if the authors could expand the introduction section to include more information and background related to cultural factors that impact healthcare seeking of women experiencing breast cancer symptoms. Some of this literature could be compared to other cultures to give the reader more of an idea how the health care seeking behavior in Ethiopia is different or similar to other countries/cultures

2. Most up-to-date data sets should be used throughout the paper. 1st paragraph of the background should use most recent GLOBOCAN data (2018 Factsheet)-cite specific figures.

3. “Breast cancer is the most prevalent cancer in African women, although its incidence is lower compared to high-income countries”. Please provide evidence for this statement.

4. It would be good to have information regarding breast cancer’s incidence and mortality rates in Ethiopia and then compare it other developed countries to show the disproportionate burden of the disease.

5. “In Ethiopia, According to the Addis Ababa……...”, please change ‘’According’’ to according.

6. “Advanced stage and large tumour size at diagnosis are associated with decreased survival”. Please cite evidence for this statement.

7. …….. “A study conducted in 2011 among breast cancer patients in Ethiopia also revealed 47.8% of study participants were nothing know about breast cancer and never heard of the - 5 - disease at all [14]” Please make this claim clear to readers.

8. It would be very effective in addressing delay presentations if authors also compared developed western health care systems that have recently reported on presentations of breast cancer. It is also necessary to include information about the current breast cancer screening guidelines in Ethiopia. Because decreasing delay presentation and breast cancer mortality rate is also dependant on the availability of appropriate screening and timely treatment of breast cancer, some information about the current screening, diagnosis and treatment of breast cancer in Ethiopia would be very helpful here as well.

Methods

Study area and approach

The study setting is well described, however, there is no mention on the study design. Authors are advised to clearly state the study’s design, the rational for choosing the design whiles citing evidence to support it.

Participants of the in-depth interview and recruitment procedures

This section needs further description to aid in future transferability and replication of the study. The target population has been specified but the criteria for inclusion and exclusion are missing. Please add them. The participants recruitment procedure has not been described. Please provide detail description of how the study participants were recruited and enrolled into the study.

Data Collection Procedure

1. How were the participants sampled? Please include this.

2. Please what informed the selection of six and eight patients from the respective hospitals involved in the study

3. Please how did the authors determined patients with serious illness and communication difficulty. It will be very advisable for authors to clearly specify the eligibility criteria.

4. The interview guide was developed through literature review. Please describe the development process. How did the authors ascertained its reliability (thus, the ability to answer the study objective(s)? Were the guiding questions piloted? If yes, what was the outcome? Please include a type of semi-structured questions asked.

5. Please indicate the experience/expertise of researcher who conducted qualitative data collection.

6. There is no mention on how the participants were contacted. Who did the first contact? How were the interview appointments scheduled? Date, time, venue

7. Please calculate the average interview time.

8. What was the interview language?

9. How did the authors determined that the information has reached saturation?

10. Please describe how participants emotional reactions were addressed in the course of sharing their experiences.

11. It would be very helpful for the authors to provide detail on how data saturation was achieved since two individuals collected the data. Please indicate the number of interviews conducted by each interviewer.

12. Did each interviewer focused on one hospital or they conducted the interviews across the hospitals involved in the study.

13. Please describe how the anonymity and confidentiality were maintained in the data collection process.

14. Please clarify whether there was any change to the interview guide as the authors went along; often in qualitative methodologies, there is an iterative process where the questions are subsequently added or shaped based on the data that emerges prior to determining saturation was achieved.

15. How was the data managed?

Data Analysis procedure

1. Please who transcribed the data? Interviewers or different people?

2. Verbatim transcription in what language. Please describe the process of the translation into English and the background of the translators.

3. What analysis technique was employed for the data analysis. There is inconsistency on data analysis technique in the abstract and in this section. Please align. Authors need to describe specifically how they conduct analysis of this study.

16. Who analyzed the data? Please indicate the experience/expertise of researcher who conducted qualitative data analysis.

17. I suggest a reference to the open code software version 4.02. Please describe the analysis process sufficient detailed, so the readers have a clear understanding of how the analysis was carried out. The analysis is not clearly described i.e how text was categorized and developed to themes. Please, give some more details about the data analysis.

18. There is an obvious bias that is integrated into the qualitative analysis if interviewee and data analyst differ. The authors should explicitly explain whether and how this was addressed in their qualitative investigation, and speculate as to how this may have biased the results. They may also want to comment on whether both men and women were involved in the interpretation of findings as the research specifically focuses on women.

Ethics consideration

Please provide the ethical approval number

Rigor/trustworthiness

There is no mention on the study rigor and this is a major flaw of the study. How did the authors ensured the trustworthiness of this qualitative study?

Result

1. Under the data collection procedure, the authors claim they conducted in-depth interview on 14 newly diagnosed breast cancer patients who had self-reported delay of more than three month from their initial symptom recognition until their first health care visit. Now, in the results section, the authors report that majority of the patients have delay of more than six month from their initial visit. Please this is a confusing conflicting statement indicative of inconsistency and flaw. Please address this.

2. It would be easier for readers to appreciate the findings if authors specify the main themes and their sub-themes. Putting them in a table may be helpful. Table 2 is depicting the codes assigned to the recurring phrases from the participants’ narrations and not the main findings (main theme and sub-themes).

3. The result is described with a several quotations. Some of the quotations stands alone, so it would therefore facilitate for the reader if you could insert some short sentences in between the quotations to connect the paragraphs with each other.

4. In the "themes" section, there are quotes that do not demonstrate the claim/point made. Authors need to look closely to ensure appropriate quotes are used.

5. When there are more citations than authorial text, the analysis is usually incomplete, which I find in this paper. Why are only a few participants quotes included?

6. These results do not differ significantly from the general late presentation of breast cancer literature. They do, however, highlight some patient-related mediators, some of which may be difficult to understand without clarity around the early detection practices and why women would wait for over six months after symptom discovery and appraisal. The novel contributions of this study may be strengthened by providing further background around early detection and diagnostic practices in Ethiopia (e.g. lack of screening facilities, lack meaningful diagnostic facilities, lack of aid from social networks), or on specifically discussing in the discussion how poverty, stigma, fear and limited access to quality care have directly affected these women.

7. The author's results may be easier to interpret if they begin their results section in commenting on the demanding patient mediated factors that is highlighted in the results under the main themes and sub-themes. As it stands now, it seems the main themes had no sub-themes.

Discussion:

As a reader, I am somewhat surprised that all participants had similar experiences and that there are no specific experiences that may have differed between participants. This makes me wonder if the sample was not generalizable, or if these themes do truly apply to all women presenting late with breast cancer symptoms in Ethiopia. This should be discussed and explicitly addressed with the author's reflections

A clinical implications section would be helpful to help guide healthcare workers who are interacting with this population.

Language

Please look through your language and make improvements, for instance long sentences. Attention should also be paid to editing as there are some grammatical errors throughout the paper.

Thank you.

6. PLOS authors have the option to publish the peer review history of their article (what does this mean?). If published, this will include your full peer review and any attached files.

Reviewer #1: **Yes: **Shahid Mahmood Baig, Professor, Head of Health Biotechnology and Group Leader of Human Molecular Genetics at National Institute for Biotechnology and Genetic Engineering (NIBGE), Faisalabad 38000, PAKISTAN

Reviewer #2: **Yes: **ADWOA BEMAH Boamah Mensah

---

## [Author Response · Author response to Decision Letter 0]

30 Oct 2020

Dear, Editor,

Greetings!

Thank you very much for all the comments provided regarding our manuscript entitled Why women with breast cancer presented late to health care facility in North-western Ethiopia? A qualitative study which are fully accepted and included in the revised version. I have accordingly made necessary revisions on the paper following the comments provided from the reviewers and editor. I also attached the themes and subthemes as supplementary file 2 and in-depth interview guide as supplementary file 1 for the reviewers based on the request. For your kind consideration, please find a point by point response to the comments in the next page of this letter and a submitted new revised version of the manuscript. 

All new changes have been highlighted in dark blue in the main document in order to facilitate review.

I hope that you will find the edits as per your expectation and I look forwards to hear from you

soon.

Yours Sincerely,

 Aragaw Tesfaw (MPH, Epidemiology)

Lecturer, Department of Public health 

College of health sciences, Debre Tabor University 

Email: aragetesfa05@gmail.com

Phone: 251921743820

1. Point by point responses to editor comments 

 Author Reponses: corrected /edited based on the given editor comment 

2. Point by point responses to the comments for reviewers 

Comments to the Author

1. Is the manuscript technically sound, and do the data support the conclusions?

Reviewer #1: Partly

Reviewer #2: No

Author Reponses: we tried to modify the manuscript based on the findings.

 2. Has the statistical analysis been performed appropriately and rigorously? 

Reviewer #1: No

Reviewer #2: N/A

Author Reponses: since the study is a qualitative study, statistical analysis is not applicable 

3. Have the authors made all data underlying the findings in their manuscript fully available?

Reviewer #1: No

Reviewer #2: No

Author Reponses: We put all relevant data within the paper and its Supporting Information files 1&2. 

4. Is the manuscript presented in an intelligible fashion and written in Standard English?

Reviewer #1: No

Reviewer #2: No

Author Reponses: corrected/edited 

 Review Comments to the Author

3. Reviewer #1: It is a very good effort to describe the reasons underlying late presentation of breast cancer in North Western Ethiopia. Some genuine reasons have been presented as to why the breast cancer patients approach late to the health care facilities in that part of the country. However, the data of only 14 breast cancer patients spanned over the record of 1-2 months is not insufficient for any statistically significant information. I would suggest the authors to include as much number of patients as possible to report a significant data for such type of study.

Author Reponses: We appreciate the reviewer comments but we determined the sample size based on the theoretical saturation point of data during data collection time when recurrent patterns become evident in the patients’ narrations or when no new data is emerged after the interviews of 14 participants. However, we planned to explore the barriers for breast cancer early diagnosis by including focus group discussions and in-depth interviews from health care providers, community health agents (locally called health development armies in Ethiopia), religious leaders, and from family members and communities perspective in another study. But in this study we just focused on patient’s perspective only, so the data collection was stopped when there was new data emerged from patient’s narrations (saturation of data). 

4. Reviewer #2: Background:

1. This is a qualitative research design that examined the experiences of 14 women who delayed seeking medical care for breast cancer symptoms and were diagnosed with advanced breast cancer. I think the manuscript would be more interesting if the authors could expand the introduction section to include more information and background related to cultural factors that impact healthcare seeking of women experiencing breast cancer symptoms. Some of this literature could be compared to other cultures to give the reader more of an idea how the health care seeking behavior in Ethiopia is different or similar to other countries/cultures.

Author Reponses: corrected and edited based on the reviewer comments with the addition of some related literatures. 

2. Most up-to-date data sets should be used throughout the paper. 1st paragraph of the background should use most recent GLOBOCAN data (2018 Factsheet)-cite specific figures.

Author Reponses: corrected/edited based on the reviewer comments 

3. “Breast cancer is the most prevalent cancer in African women, although its incidence is lower compared to high-income countries”. Please provide evidence for this statement.

Author Reponses: corrected/ edited 

5. It would be good to have information regarding breast cancer’s incidence and mortality rates in Ethiopia and then compare it other developed countries to show the disproportionate burden of the disease.

Author Reponses: corrected/ edited 

5. “In Ethiopia, According to the Addis Ababa……...” please change ‘’According’’ to according.

Author Reponses: corrected/ edited 

6“Advanced stage and large tumor size at diagnosis are associated with decreased survival”. Please cite evidence for this statement.

Author Reponses: corrected/ edited 

6. …….. “A study conducted in 2011 among breast cancer patients in Ethiopia also revealed 47.8% of study participants were nothing know about breast cancer and never heard of the - 5 - disease at all [14]” Please make this claim clear to readers.

Author Reponses: corrected/ edited 

8. It would be very effective in addressing delay presentations if authors also compared developed western health care systems that have recently reported on presentations of breast cancer. It is also necessary to include information about the current breast cancer screening guidelines in Ethiopia. Because decreasing delay presentation and breast cancer mortality rate is also dependent on the availability of appropriate screening and timely treatment of breast cancer, some information about the current screening, diagnosis and treatment of breast cancer in Ethiopia would be very helpful here as well.

Author Reponses: corrected/ edited 

Methods

Study area and approach

the study setting is well described, however, there is no mention on the study design. Authors are advised to clearly state the study’s design, the rational for choosing the design whiles citing evidence to support it.

Author Reponses: corrected/ edited in the main document: our study design is a phenomenological study 

Participants of the in-depth interview and recruitment procedures. This section needs further description to aid in future transferability and replication of the study. The target population has been specified but the criteria for inclusion and exclusion are missing. Please add them. The participant’s recruitment procedure has not been described. Please provide detail description of how the study participants were recruited and enrolled into the study.

Author Reponses: Corrected/ edited based on the comment 

Data Collection Procedure

1. How were the participants sampled? Please include this.

Author Reponses: The participants in this study were newly diagnosed female breast cancer patients in the two specialized hospitals (University of Gondar and Felege Hiwot Specialized hospitals) and they were selected using purposeful sampling technique.

2. Please what informed the selection of six and eight patients from the respective hospitals involved in the study?

Author Reponses: Two data collectors with the assistance of two note takers were assigned in the two hospitals (one data collector and one note take in each hospitals). Then the sample size was determined based on the theoretical saturation of data in each of the study sites. 

3. Please how the authors did determined patients with serious illness and communication difficulty. It will be very advisable for authors to clearly specify the eligibility criteria.

Author Reponses: corrected/ edited based on the document in the main document

4. The interview guide was developed through literature review. Please describe the development process. How the authors did ascertained its reliability (thus, the ability to answer the study objective(s)? Were the guiding questions piloted? If yes, what was the outcome? Please include a type of semi-structured questions asked.

Author Reponses: we used a semi-structure interview guide to collect data from the participants. The guide was developed by reading different literatures and it was submitted to experts in the field (clinical and surgical oncologists) and other health care providers to see judge content and face validity. It was initially prepared in English version but translated to the local language (Amharic). 

5. Please indicate the experience/expertise of researcher who conducted qualitative data collection.

Author Reponses: Two public health professionals who had a Master of public health in health education and promotion collected the data with the assistance of two note takers (cancer trained clinical nurses). They are lecturers and have experience in qualitative research. 

6. There is no mention on how the participants were contacted. Who did the first contact? How were the interview appointments scheduled? Date, time, venue

Author Reponses: Participants of the study were selected with the help of the oncology unit nurses. After we obtained informed consent to participate, an appropriate place and time for an interview was arranged by the oncology unit coordinators. Interviews were conducted face to face with participants in a private room at the oncology unit.

7. Please calculate the average interview time.

Author Reponses: Each interview had taken approximately 40-60 minutes with the average interview time of 46.67 minutes

8. What was the interview language?

Author Reponses: The local language which is called Amharic was used for in-depth interview 

9. How did the authors determined that the information has reached saturation?

Author Reponses: the theoretical saturation point of data during data collection time was used to determine the sample size required in our study. This was defined when recurrent patterns become evident in the patients’ descriptions or when no new data is arose after the interviews. 

10. Please describe how participant’s emotional reactions were addressed in the course of sharing their experiences.

Author Reponses: As the reviewer said when emotions raised during interviews from patients, the data collectors tried to reassure the patients and if necessary they link to oncologists and clinical psychologists for more reassurance. But it was not that much challenge in our study. 

11. It would be very helpful for the authors to provide detail on how data saturation was achieved since two individuals collected the data. Please indicate the number of interviews conducted by each interviewer.

Author Reponses: One data collector was assigned with one note taker in each of the hospitals. Then each interviewer focused on only on their assigned hospital and interview the breast cancer patient’s reasons for their late presentation using the interview guide and they stopped when they could not get new information’s from patient narrations for the interviews. 

12. did each interviewer focused on one hospital or they conducted the interviews across the hospitals involved in the study.

Author Reponses: Yes! Each interviewer were focused on one hospital which they were assigned and they continued interview until no new data is emerged from participants. 

13. Please describe how the anonymity and confidentiality were maintained in the data collection process.

Author Reponses: Confidentiality of information and privacy of participants’ interview was respected. The participants were told that information they provide use only for the purpose of this study. The names of the participants did not included in the interview guide rather specific codes were used. All documents were kept private and confidential. All audio-recorded interviews were reviewed by the transcriber and the principal investigator only, and each participant was identified by specific code number, rather than by name. 

14. Please clarify whether there was any change to the interview guide as the authors went along; often in qualitative methodologies, there is an iterative process where the questions are subsequently added or shaped based on the data that emerges prior to determining saturation was achieved.

Author Reponses: Yes! There was updates on the interview guide which was due to emerging of new ideas from the patients while interviewing. The interviewers used probing like questions to explore more in-depth information from the patients.

15. How was the data managed?

Author Reponses: All interviews were audio-recorded after the verbal consent of the study participants. Field notes were also taken during the in-depth interview to include keynotes of the participants and to use as a backup in case the video records are lose. There was debriefing each day after data collection with data collectors and investigators. Each collected data were also kept in password protected computer with the principal investigator. 

Data Analysis procedure.

1. Please who transcribed the data? Interviewers or different people?

Author Reponses: data was transcribed by one another master of public health student and it was again cross-checked by two other independent persons! 

2. Verbatim transcription in what language. Please describe the process of the translation into English and the background of the translators.

Author Reponses: Verbatim transcription was done first in local language/Amharic/ then translated to English. The translators were fluent in both English and Amharic language! 

3. What analysis technique was employed for the data analysis? There is inconsistency on data analysis technique in the abstract and in this section. Please align. Authors need to describe specifically how they conduct analysis of this study.

Author Reponses: We used thematic content analysis approach to analyze the data in Open Code software.

16. Who analyzed the data? Please indicate the experience/expertise of researcher who conducted qualitative data analysis.

Author Reponses: the investigators were analyzed the data by consulting other experts in the field and who had qualitative research experience. One of the author has also experience in participating and doing qualitative researches. 

17. I suggest a reference to the open code software version 4.02. Please describe the analysis process sufficient detailed, so the readers have a clear understanding of how the analysis was carried out. The analysis is not clearly described i.e. how text was categorized and developed to themes. Please, give some more details about the data analysis.

Author Reponses: Modifications are made based on the comment in the document 

18. There is an obvious bias that is integrated into the qualitative analysis if interviewee and data analyst differ. The authors should explicitly explain whether and how this was addressed in their qualitative investigation, and speculate as to how this may have biased the results. They may also want to comment on whether both men and women were involved in the interpretation of findings as the research specifically focuses on women.

Author Reponses: yes! The interviewer and data analyzer were different but there was always debriefing and discussion after data collection daily between the data collectors and investigators. The data of this qualitative research were collected only female breast cancer patients. 

Ethics consideration

Please provide the ethical approval number

Author Reponses: corrected/ edited 

Rigor/trustworthiness

there is no mention on the study rigor and this is a major flaw of the study. How the authors did ensured the trustworthiness of this qualitative study?

Author Reponses: Descriptions were made in the main document as follows: The transferability of the findings was ensured by collecting data on two oncology units which are used as referral centers for cancer patients. This helped to get women from different corners of the region with different socio-demographic and cultural background. The data collectors and investigators were met on a daily basis after data collection and debriefed on their daily interview findings. To maintain the trust worthiness and validity of the findings, the investigators developed rapport with patients who were participated in the study. The dependability of data was maintained by taking depth of information until reaching data saturation point which reflects a sample needed for the study i.e. when repeated patterns become apparent in the patients' narrations. 

Result

1. Under the data collection procedure, the authors claim they conducted in-depth interview on 14 newly diagnosed breast cancer patients who had self-reported delay of more than three month from their initial symptom recognition until their first health care visit. Now, in the results section, the authors report that majority of the patients have delay of more than six month from their initial visit. Please this is a confusing conflicting statement indicative of inconsistency and flaw. Please address this.

Author Reponses: Self-reported delay of more than three month from their initial symptom recognition until their first health care visit was our inclusion criteria to select participants but the majority had delay of more than six month so we included them to participate since our sampling technique was purposive focusing on which group of patients will give us more information on reasons for late presentation.

2. It would be easier for readers to appreciate the findings if authors specify the main themes and their sub-themes. Putting them in a table may be helpful. Table 2 is depicting the codes assigned to the recurring phrases from the participants’ narrations and not the main findings (main theme and sub-themes).

Author Reponses: corrected/ edited (See supplementary file_2) 

3. The result is described with a several quotations. Some of the quotations stands alone, so it would therefore facilitate for the reader if you could insert some short sentences in between the quotations to connect the paragraphs with each other.

Author Reponses: Modifications were made based on the reviewer comments on the main document 

4. In the "themes" section, there are quotes that do not demonstrate the claim/point made. Authors need to look closely to ensure appropriate quotes are used.

Author Reponses: corrected/ edited based on the comment 

5. When there are more citations than authorial text, the analysis is usually incomplete, which I find in this paper. Why are only a few participants quotes included?

Author Reponses: corrected/ edited 

6. These results do not differ significantly from the general late presentation of breast cancer literature. They do, however, highlight some patient-related mediators, some of which may be difficult to understand without clarity around the early detection practices and why women would wait for over six months after symptom discovery and appraisal. The novel contributions of this study may be strengthened by providing further background around early detection and diagnostic practices in Ethiopia (e.g. lack of screening facilities, lack meaningful diagnostic facilities, lack of aid from social networks), or on specifically discussing in the discussion how poverty, stigma, fear and limited access to quality care have directly affected these women.

Author Reponses: As we tried to mention in the background section, there are limited oncology centers in Ethiopia and cancer did not get focus in the country. In recent times some of referral and specialized hospitals start pathologic testing like FNC and biopsy and around six hospitals start oncology service for cancer patient’s surgery, chemotherapy. But there is only one radiotherapy center (Tikur Anbesa Hospital) in the country for more than 110 million population. 

7. The author's results may be easier to interpret if they begin their results section in commenting on the demanding patient mediated factors that is highlighted in the results under the main themes and sub-themes. As it stands now, it seems the main themes had no sub-themes.

Author Reponses: Corrected/ edited based on the comment! 

Discussion: 

As a reader, I am somewhat surprised that all participants had similar experiences and that there are no specific experiences that may have differed between participants. This makes me wonder if the sample was not generalizable, or if these themes do truly apply to all women presenting late with breast cancer symptoms in Ethiopia. This should be discussed and explicitly addressed with the author's reflections.

Author Reponses: corrected/ edited 

A clinical implications section would be helpful to help guide healthcare workers who are interacting with this population.

Author Reponses: We write the clinical implications of this study findings!

Language

Please look through your language and make improvements, for instance long sentences. Attention should also be paid to editing as there are some grammatical errors throughout the paper.

Thank you.

Author Reponses: modifications on the grammar were made based on the comments 

6. PLOS authors have the option to publish the peer review history of their article (what does this mean?). If published, this will include your full peer review and any attached files.

Do you want your identity to be public for this peer review? For information about this choice, including consent withdrawal, please see our Privacy Policy.

7. Reviewer #1: Yes: Shahid Mahmood Baig, Professor, Head of Health Biotechnology and Group Leader of Human Molecular Genetics at National Institute for Biotechnology and Genetic Engineering (NIBGE), Faisalabad 38000, PAKISTAN

8. Reviewer #2: Yes: ADWOA BEMAH Boamah Mensah

---

## [Decision Letter · Decision Letter 1]

24 Nov 2020

Why women with breast cancer presented late to health care facility in North-western Ethiopia? A qualitative study

PONE-D-20-18576R1

Dear Dr. Tesfaw,

We’re pleased to inform you that your manuscript has been judged scientifically suitable for publication and will be formally accepted for publication once it meets all outstanding technical requirements.

Kind regards,

Alvaro Galli

Academic Editor

PLOS ONE

Additional Editor Comments (optional):

Reviewers' comments:

Reviewer's Responses to Questions

**Comments to the Author**

1. If the authors have adequately addressed your comments raised in a previous round of review and you feel that this manuscript is now acceptable for publication, you may indicate that here to bypass the “Comments to the Author” section, enter your conflict of interest statement in the “Confidential to Editor” section, and submit your "Accept" recommendation.

Reviewer #1: All comments have been addressed

2. Is the manuscript technically sound, and do the data support the conclusions?

Reviewer #1: Partly

3. Has the statistical analysis been performed appropriately and rigorously? 

Reviewer #1: N/A

4. Have the authors made all data underlying the findings in their manuscript fully available?

Reviewer #1: Yes

5. Is the manuscript presented in an intelligible fashion and written in standard English?

Reviewer #1: Yes

6. Review Comments to the Author

Reviewer #1: The authors have addressed all the comments adequately. The manuscript is recommended in its present form provided a very careful proof reading is done.

7. PLOS authors have the option to publish the peer review history of their article (what does this mean?). If published, this will include your full peer review and any attached files.

Reviewer #1: **Yes: **Shahid Mahmood Baig

---

## [Editor Report · Acceptance letter]

27 Nov 2020

PONE-D-20-18576R1 

Why women with breast cancer presented late to health care facility in North-west Ethiopia?  A qualitative study 

Dear Dr. Tesfaw:

I'm pleased to inform you that your manuscript has been deemed suitable for publication in PLOS ONE. Congratulations! Your manuscript is now with our production department. 

Kind regards, 

on behalf of

Dr. Alvaro Galli 

Academic Editor

PLOS ONE